# Simultaneous Removal of Cyanide and Heavy Metals Using Photoelectrocoagulation

**Ahmad Shahedi [1], Ahmad Khodadadi Darban [2,\*], Ahmad Jamshidi-Zanjani [2,\*], Fariborz Taghipour [3] and Mehdi Homaee [2]**

1   Department of Mineral Processing, Faculty of Engineering, Tarbiat Modares University, Tehran 14115, Iran
2   Department of Mining and Environmental Engineering, Faculty of Engineering, Tarbiat Modares University, Tehran 14115, Iran
3   Department of Chemical and Biological Engineering, University of British Columbia, Vancouver, BC V6T1Z3, Canada
\*   Correspondence: akdarban@modares.ac.ir (A.K.D.); ajamshidi@modares.ac.ir (A.J.-Z.)

**Abstract:** One of the new methods used to remove the contaminants from effluent is the electrocoagulation method, which is sometimes combined with other methods to increase the removal efficiency of contaminants. To simultaneously remove nickel, cyanide, zinc, and copper, the combined method of photo-electrocoagulation was used along with an oxidizing agent, namely hydrogen peroxide (Hp). In addition, the effects of factors affecting the removal efficiency were studied, including pH, electrode arrangement, and current intensity. An electric current of 300 mA at a pH of 10 for 60 min, Fe-SS electrodes with a distance between them of 5 cm, and hydrogen peroxide at a rate of 4 mg/L were the ideal conditions needed to accomplish the photo-electrocoagulation-oxidation process. According to these study findings, when the combined method of photocatalyst-electrocoagulation-oxidation (Hp) was used, the highest removal efficiencies of nickel, cyanide, zinc, and copper were 85, 96, 94, and 98%, respectively. The results showed that using the combined photo-electrocoagulation-oxidation method increased the efficiency of simultaneous removal of pollutants by 10% compared to conventional electrocoagulation method. The reason for the increase in removal efficiency is the production of hydroxyl radicals simultaneously with the formation of coagulants produced by electrocoagulation process.

**Keywords:** zinc; nickel; copper; cyanide; photo-electrocoagulation

## 1. Introduction

Heavy metals including lead, copper, zinc, chromium, mercury, nickel, and cadmium are among the toxic contaminants in the surface and groundwater, which enter the environment mainly through various industrial and mining effluents, including plating. Mining operations are one of the most frequent sources of pollution in aquatic habitats. The creation of effluents containing various kinds of heavy metals and dangerous substances, including cyanide, is only one of the latest environmental issues brought on by mining and associated businesses in addition to environmental deterioration. Mine effluents contain various heavy metals, such as iron, zinc, copper, manganese, and chromium; however, in most cases, they contain cyanide, carbonate, sulfate, arsenic, and their complexes [1]. Furthermore, given the high mobility of heavy metals, they are likely to be widely distributed in the aquatic ecosystems and cause many environmental problems for humans due to their toxicity and ability to form complexes with higher toxicity [2]. Effluents produced by various industries, including plating, metal processing such as gold/silver processing, steel/pesticide/pharmaceutical manufacturing, and oil refining, contain large amounts of toxic ions of cyanide, zinc, nickel, copper, etc. [3]. Cyanide species are generally toxic and could be dangerous to human health even at low concentrations; they could form weak and strong cyanide-metal complexes if present in effluents. Nickel- and copper-cyanide are

strong complexes, and zinc-cyanide is one of the weak complexes. Toxicity and stability of strong complexes are high, while weak complexes have low toxicity and stability. Thus, these effluents should be completely treated before discharging to surface and groundwater [4]. According to a standard provided by the US Environmental Protection Agency (USEPA), the permissible limit of cyanide in drinking water is up to 200 ppb. In addition, the permissible limits of copper and zinc in water are 1.3 and 1 mg/L, respectively [5]. Adsorption, precipitation, reverse osmosis, and ion exchange are some of the typical techniques used to remove heavy metals and hazardous components in wastewater [6]. One of the major issues with using standard techniques is that they cannot concurrently remove cyanide, heavy metals, and existing complexes [6]. Other challenges related to using the conventional and physical methods, such as precipitation, include the production of large amounts of sludge as well as the use of coagulant chemicals, challenging their usage. Filter membrane clogging, treatment costs, and treatment limitations are some of the factors challenging the use of other methods. One of the methods, which were considered by researchers, is the electrochemical process. The reason for using this method is its many advantages over other conventional methods, including the elimination of a wide range of toxic pollutants and the production of less sludge [7]. This method is divided into subcategories, such as electrical flotation, electrical coagulation, and electrical deposition [8]. In the electrocoagulation method (EC), using an electric current to the anode and cathode electrodes, the sacrificial anode, which is mainly made of iron, is oxidized and produces a coagulant solution that could trap a so-called broom contaminant. In the solution after coagulation, finally, the coagulants are attached to trapped contaminants.

Regarding the relatively low efficiency of electrocoagulation method in the simultaneous removal of contaminants, so far different methods have been combined with this method to increase its efficiency and remove as many different contaminants as possible, such as different metal complexes and resistant contaminants. One of the methods, which was combined with electrocoagulation method and has led to an increase in EC efficiency and a decrease in the sludge production, is the advanced oxidation method. In this method, photo-catalysts, hydrogen peroxide, ozone, and Fenton's reagent are used to produce highly reactive hydroxyl radicals ($OH^-$). Under typical temperature and pressure settings, enough OH radicals are produced during the wastewater treatment process to hasten the degradation of organic hazardous contaminants. Hydrogen peroxide ($H_2O_2$) might oxidize and remove more contaminants in the effluent by converting into hydroxyl radicals ($OH-$). The reason is the higher oxidation-reduction potential of OH radicals compared to other oxidants (2.76 volts) [9]. Recent research showed that using the photo-catalysts such as UV-LED to remove organic pollutants from the effluent has many benefits. The sources of photo-catalysts are UV-LED lamps that are used to assess the efficiency of photo-catalysts. UV lamps are available in three different wavelengths and accordingly classified, including UVC (180–280 nm, 8 watts), UVB (280–320 nm, 11 W), and UVA (320–420 nm, 4 W) [10]. Consequently, the combined photo-electrocoagulation method with hydrogen peroxide (Hp) was used to eliminate contaminants such as cyanide and heavy metals. So far, several studies have been performed to remove toxic compounds such as cyanide and heavy metals. Fu et al. (2009) conducted a study to eliminate Ni (II) from Ni EDTA effluent using a combination of Fenton and precipitation (hydroxide precipitation) methods. According to the findings, more than 92% of Ni (II) was eliminated within 60 min at pH 3.0 during the Fenton process and at pH 11.0 during the precipitation process. The initial concentration of hydrogen peroxide and ferric ions was 141 and 1 mmol, respectively [11]. Kim et al. (2018) also conducted a study to eliminate cyanide from the effluent using UV-LED/$H_2O_2$/$Cu^{2+}$ system, the results showed that complete removal of cyanide from the effluent was achieved within 30 min at pH = 11 [12]. For simultaneous removal of lead and cyanide from effluents generated by mineral processing plants, Kohzadi et al. (2021) performed several tests using electrocoagulation with iron-aluminum electrodes. According to the findings, the maximum removal efficiencies of cyanide (97%) and lead (81%) were obtained at pH 9 with an electric current of 300 mA [13]. Kobya et al. (2017) conducted a study

to eliminate zinc-cyanide from effluents produced by the plating industry using the EC method. The results showed that the removal of more than 99% of zinc-cyanide was achieved within 60 min at pH = 9.5 and a current density of 60 A/m$^2$ [14]. Akbal et al. (2011) also performed a study to remove heavy metals from plating effluents using the EC method. The results showed that Ni, Cr, and Cu were completely removed from the effluents at pH = 9.0 and a current density of 10 mA/cm$^2$ using iron-aluminum as an anode-cathode electrode arrangement [15].

Zhou et al. (2022) used a layered cationic framework material loaded with the phosphonate as an intermediate to adsorb hazardous metals from an aqueous solution. The results showed that double-layer hydroxide removed $Zn^{2+}$ and $Fe^{3+}$ pollutants from water due to its excellent chelation absorption property. Moreover, the results showed that the synthesized material has great potential for purifying toxic pollutants [16]. In another research conducted by Zhou et al. (2022) by the purpose of adsorbing heavy metal ions from the solution, functionalized double-layered phosphonate hydroxide was used. The results showed that the functionalized composite has a high potential to adsorb heavy metals, so that its use led to the maximum adsorption capacity of 156.95 mg/g ($Cr^{3+}$) and 198.34 mg/g ($Cd^{2+}$) separately. The adsorbent can be used after six cycles; it was satisfactorily reusable [17].

In the experiments conducted in previous research, the effect of other pollutants on the removal of the studied elements was not deeply considered. Due to the presence of elements, such as zinc, copper, and nickel in the effluent of gold processing plants, the possibility of forming strong and weak cyanide-metal complexes (WAD-SAD) in the effluent is high. These complexes compete with other pollutants and prevent the removal of heavy metals in the effluent, thereby reducing their removal efficiency.

Conventional techniques lack the necessary efficacy to get rid of these contaminants. It is thought that there is a research gap since this subject has not received as much attention in earlier studies. The pH needed to remove cyanide is usually neutral to alkaline, but the pH needed to remove heavy metals is mostly acidic. Considering the mismatch among the pHs required to remove heavy metals and cyanide separately, how to simultaneously eliminate heavy metals and cyanide by bringing together the various pHs required is another research gap not addressed in previous research. Conventional methods do not show a relatively good performance in removing cyanide and heavy metals simultaneously, and the efficiency of simultaneous elimination of these contaminants has been reported to be low in existing studies. The main novelty of this study is using a combination of conventional and novel methods (electrocoagulation-photocatalyst-oxidizing agent) to increase the efficiency of simultaneous elimination of heavy metals and cyanide from mining effluents. The mechanism of this combined method was investigated. Another aspect of this research is its attempt to bring together the various pHs required to simultaneously remove heavy metals and cyanide. Moreover, this research aimed to simultaneously eliminate heavy metals and cyanide from synthetic effluents by considering different operating conditions, as well as to evaluate the performance of the combined photo-electrocoagulation-oxidation method in removing heavy metals (Pb-Zn-Ni) and cyanide simultaneously from synthetic effluents.

The novelty of the present research is the optimization of simultaneous removal of cyanide and metal ions of copper, zinc, and nickel from synthetic wastewater. Furthermore, considering the pH of removing these pollutants is in different ranges, finding the optimal pH in which the maximum and simultaneous removal of pollutants occurs was another novelty. Furthermore, the effectiveness of combined method (electrocoagulation-photocatalyst-oxidizer) for the simultaneous removal of the studied pollutants from the real effluent of the gold processing plant using the enhanced electrocoagulation was investigated, which was also among the cases that were rarely addressed in the previous studies.

## 2. Materials and Methods

### 2.1. Setup and Instruments

The effluent was used to perform electrocoagulation experiments. Cyanide, copper, and zinc were used to simulate synthetic effluent to real mine effluents. The concentrations of cyanide and heavy metals (zinc, copper, and nickel) used to produce synthetic effluent were 198 and 168 mg/L, respectively. The type and amount of chemicals added were selected based on the previously published studies, as well as the analysis of real mine effluents collected from an active gold mine in Yazd province, Iran [18]. The specifications of the synthetic effluent produced for use in EC experiments are presented in detail in Table 1.

**Table 1.** Specifications of the effluent prepared for use in EC experiments.

| As (ppb) | Ca (ppm) | CN (ppb) | Hg (ppb) | Ni (ppm) | Mg (ppm) | Zn (ppm) | Cu (ppm) | pH |
|---|---|---|---|---|---|---|---|---|
| 81.88 | 240.2 | 2850 | 102.09 | 0.209 | 13.01 | 0.01 | 0.661 | 6.68 |

To prepare synthetic effluent similar to gold mine effluents, three types of chemical compounds were included in the effluent. Sodium cyanide (NaCN, $\geq$98.0%), nickel (II) sulfate (NiSO$_4$. 98%), copper sulfate (CuSO$_4$.5 H$_2$O: $\geq$99.0%), and zinc sulfate heptahydrate (ZnSO$_4$.5 H$_2$O: $\geq$99.0%) were purchased from Merck. Sodium hydroxide (NaOH, Chemi-pharma) and sodium chloride (NaCl, Chemi-pharma) were used to adjust solution pH and create an alkaline environment, respectively. The required volume of effluent solution was prepared using deionized water. The total volume of synthetic effluent prepared with the specified concentrations of cyanide, copper, and zinc and used in experiments was 2 L. Iron (Fe), aluminum (Al), and stainless steel (SS) electrodes with a purity of 90% and dimensions of $12 \times 10 \times 0.1$ cm were used. The batch-style Plexiglas electrocoagulation reactor utilized in this study has a capacity of 3 L (Figure 1). The employed electrodes were wired in a unipolar mode to a direct current (DC) power source. A power supply device delivered constant DC (SANJESH TEK, 8051). The total area and the area of the operational part of each sacrificial anode were 120 and 80 cm$^2$, respectively. When direct current was applied to the electrodes, the reaction began and ended after a specified time. Aeration was performed using an aeration pump (Aqua AP-320) at a rate of 1.5 mL/min to mix the reactor contents and increase the oxidation rate. Sampling of solution inside the reactor was performed at specified times. After filtering the suspensions by 0.45 μm pore-size paper filters, the residues on the filters were examined for copper, zinc, and cyanide content. Additionally, 1 N HCl and distilled water were used to remove the passive surface layer created on the electrodes. Furthermore, by reviewing the literature, it was found that the use of 4 mg/L of hydrogen peroxide (Hp) as an oxidizing agent could also accelerate the process of heavy metal removal. Moreover, the arrangement of the studied electrodes was examined, and the electrode that caused the maximum removal of heavy metals and cyanide separately under the provided conditions was selected for use in other experiments. Initial solution pH, reaction duration, electrode material, and current intensity were among the parameters examined in this research. Iron-aluminum was explored as an anode-cathode electrode configuration and vice versa. All experiments were run in batch mode. Investigations were also conducted on the use of stainless steel electrodes as oxidizing agents. According to previous research, the reason to choose the mentioned electrodes was their advantages such as low cost, easy access, as well as the possibility of using them as effective oxidants in the process [19]. In addition, in order to investigate the effect of advanced oxidation methods, LED (30 w) and UV (280 nm, 8 watts) lamps were used both separately and in combination with an oxidizing agent (hydrogen peroxide) in the EC method. UV lamp used in this study was UVC (Ultraviolet C), which was used according to previous research [20]. In order to obtain maximum efficiency under different conditions during the integrated EC-UV-LED process, aeration was performed by the mentioned aeration pump.

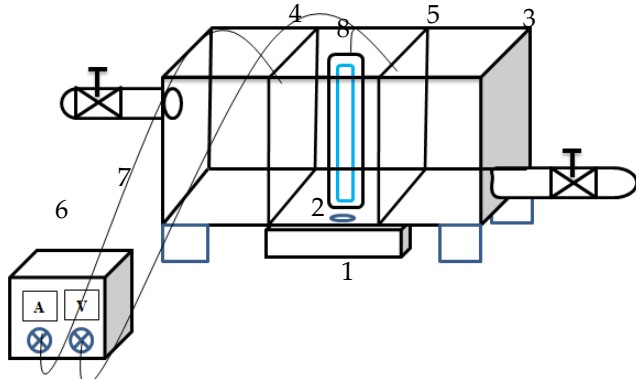

**Figure 1.** Schematic representation of an electrocoagulation reactor. 1: stirrer, 2: magnet, 3: ECP cell, 4: anode, 5: cathode, 6: power supply, 7: wires, 8: UV-LED chipset (280 nm).

*2.2. Experiments*

The simultaneous removal of cyanide and heavy metals from the effluent was investigated using several methods as follows: oxidation (hydrogen peroxide) (Hp), conventional electrocoagulation (EC), photo-electrocoagulation (PEC), and photo-electrocoagulation-oxidation (PECHp). In total, 50 experiments were performed. To investigate the influence of pH on the removal efficiency of cyanide, copper, zinc, and nickel using the oxidation method, experiments were performed at pH = 8, 9, 10, and 12 for 60 min using 4 mg/L of hydrogen peroxide. In this test, the purpose was to separate the precipitates from the coagulants by changing pH. Considering the presence of cyanide in the effluent, pH values in the neutral to alkaline range (8–12) were selected to prevent the formation of toxic hydrogen-cyanide gas (HCN). To study the effect of pH on the removal efficiency of cyanide and heavy metals using the electrocoagulation method in batch mode, experiments were performed for 60 min with an electric current of 300 mA. Electrode material is an important factor playing a significant role in the simultaneous removal of cyanide, copper, nickel, and zinc using the EC method. According to studies, using iron electrode as a sacrificial anode effectively removes soluble pollutants. As a result, in all tests, iron electrodes were used as the sacrificial anodes and Al and SS electrodes as the cathodes to simultaneously remove nickel, cyanide, zinc, and copper from the effluent using the EC process in batch mode. The most suitable pH for this test was determined to be pH = 10 according to a previous study by El-Ashtoukhy et al. (2010) [21]. Moreover, the effects of current intensity (100–200–300 mA), reaction time (10 to 60 min), and pH (8 to 12) at different levels were evaluated on the efficiency of electrocoagulation method in removing heavy metals and cyanide simultaneously. Studies showed that increasing the intensity of the electric current applied to the sacrificial anode leads to the production of more coagulant ions, their growth, and the deposition of contaminants [22]. Moreover, to study the factors affecting (current intensity, electrode material, reaction time, and pH) the performance and efficiency of EC-UV-LED method in removing cyanide and heavy metals simultaneously, the batch operation was studied. To study the performance of combined photo-electrocoagulation-oxidation (Hp) method in removing zinc, copper, nickel, and cyanide simultaneously, the effect of UV-LED was first separately studied. Based on the findings of the previous tests, the most suitable operating conditions were provided for this test to prevent testing with similar conditions. Consequently, pH 10, which was the most appropriate pH in similar tests, was considered for this test. Thus, iron and stainless-steel electrodes were utilized as the sacrificial anode and cathode, respectively, in terms of their positive results in previous similar tests. This experiment was carried out for 60 min at a current intensity of 300 mA in the presence of hydrogen peroxide and UVC-LED photo-catalysts. Hydrogen peroxide was injected into the reactor as an oxidizing agent at an optimal rate of 4 mg/L. The operating conditions studied and the levels of each are presented in Table 2.

**Table 2.** Operational conditions used to remove cyanide and heavy metals.

| Experiment | Pollutants | Operational Conditions | | | | |
|---|---|---|---|---|---|---|
| | | Mode | Current Intensity (mA) | Electrode (Anode–Cathode) | pH | Time (min) |
| Hp | | | - | - | 8–9–10–11–12 | |
| | | | 300 | Fe-SS | 8–9–10–11–12 | |
| EC | | | 300 | Fe-SS, Fe-Al | 10 | |
| | Cu-Zn- | Batch | 100–200–300 | Fe-SS, Fe-Al | 8–9–10–11–12 | 60 |
| PEC | CN-Ni | | 100–200–300 | Fe–Al, Fe-SS    UV-LED | 8–9–10–11–12 | |
| (PEC)$_{pH}$ * | | | 300 | Fe-SS    UV-LED | 8–9–10–11–12 | |
| (PEC)$_{M}$ ** | | | 300 | Fe-SS, Fe-Al    UV-LED | 10 | |
| (PEC)$_{CI}$ *** | | | 100–200–300 | Fe-SS    UV-LED | 10 | |
| PECHp | | | 300 | Fe-SS    UV-LED | 10 | |

(PEC)$_{pH}$ *: Use photo-electrocoagulation (PEC) at different pH; (PEC)$_{M}$ **: Use Photo-electrocoagulation (PEC) with electrodes of different materials; (PEC)$_{CI}$ ***: Use photo-electrocoagulation (PEC) with different current intensities.

ICP (inductively coupled plasma) spectrometry (Nexion 3300, Perkin-Elmer, Waltham, MA, USA) was used to measure the concentrations of the studied heavy metals (Cu$^{2+}$, Zn$^{2+}$, Ni$^{2+}$). To measure the cyanide content in the effluent based on the standard methods, standard titration of AgNO$_3$ solution was performed [13]. The solution pH was determined using a pH meter (Orion 920). Sampled suspensions were filtered using a paper filter with a pore size of 0.45 μm.

## 3. Results and Discussion

### 3.1. Oxidation Method (Using Hydrogen Peroxide)

Effect of Initial pH

To differentiate coagulants caused by electrocoagulation process from precipitates in terms of sedimentation process, the influence of pH was first investigated separately on the removal of each contaminant. The main difficulty in removing cyanide and heavy metals at the same time is pH mismatch. The ideal pH for the removal of cyanide is mostly alkaline, whereas the ideal pH for the removal of heavy metals is primarily acidic. As mentioned earlier, pH values in the range of 8 to 12 were used to prevent the formation of toxic HCN gas.

The effect of initial pH on the removal of contaminants using the oxidation method (hydrogen peroxide) is shown in Figure 2. As shown in Figure 2, in the oxidation process using hydrogen peroxide, the concentration of cyanide decreased from 61 to 2 mg/L with increasing pH from 8 to 12. The same trend was observed for the elements copper, zinc, and nickel, so that with the increase of pH from 8 to 10, the residual concentration of the copper decreased from 59 to 45, zinc from 63 to 46, and nickel from 111 to 94 mg/L. However, by increasing pH from 10 to 12, the residual concentration of copper increased from 45 to 58, nickel from 94 to 119, and zinc from 46 to 80 mg/L. The maximum removal efficiencies of nickel, zinc, and copper were 43, 72, and 73% at pH = 10. Moreover, the amount of cyanide removal from the solution increased with increasing pH from neutral to alkaline (pH 12), and the maximum cyanide removal efficiency of 98% was obtained at pH = 12. This is because by increasing pH the hydrolysis of metals and the formation of metal-hydroxyl complexes, such as ferric hydroxide, increase. With increasing pH, the degradation of H$_2$O$_2$ intensifies, leading to the production of metal hydroxides and thus a reduction in the amount of metal cations in the effluent [23]. These metal hydroxides (Fe(OH)$_3$) cause trapping and so-called sweeping of other metal cations and their precipitation in the effluent. However, pH values above 9 reduce the decomposition rate of H2 O2; thus, the formation of ferric hydroxides is reduced, and so is the removal efficiency of metal cations. When hydrogen peroxide is added to a cyanide-containing effluent, cyanide is oxidized to cyanate at high pHs based on the following equation [24]:

$$CN^- + H_2O_2 \rightarrow CNO^- + HO \tag{1}$$

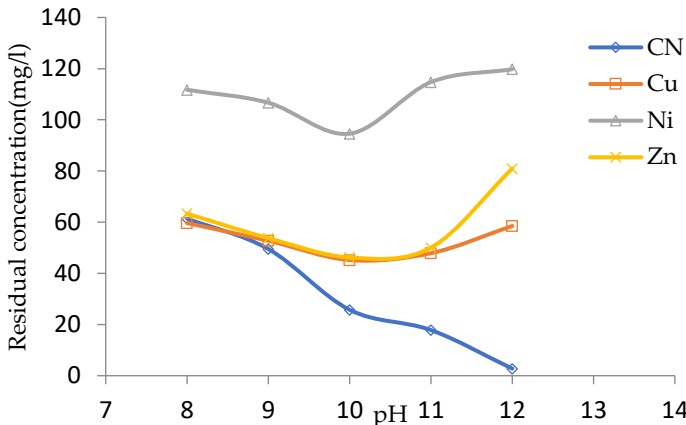

**Figure 2.** Removal of cyanide and heavy metal by (Hp) at different pHs.

Considering the alkaline conditions, cyanate is hydrolyzed to ammonium [25].

$$CNO^- + H_2O \rightarrow CO_3^{2-} + NH_4^+ \tag{2}$$

The major products obtained from cyanide oxidation include ammonium and bicarbonate. Studies showed that about 10 to 20% of cyanide is usually oxidized to ammonia during the purification process [26].

$$OCN^- + H^+ + 2H_2O \rightarrow HCO_3^- + NH_4 \tag{3}$$

By increasing the pH, the rate of $H_2O_2$ decomposition and hydroxide production increases, and in terms of overproduction of these hydroxyl radicals, the formation of soluble complexes becomes possible. As a result, as metal complexes are produced, the amount of pollutants that are still present in the effluent start to diminish. Hydroxides' electrostatic repulsion force, however, may have an impact and slow down the removal of heavy metals. Based on previous studies, the required pH for the removal of cyanide-metal complexes is 9 to 9.5, which is mainly related to the removal of iron-cyanide complex. To remove other metal-cyanide complexes, the pH of the solution must be higher, and precipitate is done via the formation of cyanide-iron-copper complex [27]. When using hydrogen peroxide, the most effective factor for removing heavy metals and their complexes is pH > 7. Sedimentation is done by metal hydroxides that trap and sweep heavy metals in effluents, as well as by cyanide-metal complexes. Thus, according to the above cases, it was found that maximum simultaneous removal of nickel, zinc, and copper using 4 mg/L of hydrogen peroxide could be achieved at pH = 9, which is attributed to the production of metal hydroxides, the adsorption of pollutants, and finally sedimentation of flocs. When hydrogen peroxide is used in the presence of heavy metals and cyanide, heavy metal ions and hydrogen peroxide react together to form heavy metal hydroxides. By adding copper sulfate to the cyanide solution, cyanide-copper (I and II) complexes are formed, which form cyanogen. Then cyanogen is combined with free cyanide and forms several compounds, such as $Cu(CN)_4^{3-}$ and $Cu(CN)_3^{2-}$. Finally, by adding hydrogen peroxide and reacting with these compounds, hydroxyl radicals are produced. The process is similar for the heavy metals zinc and nickel, which eventually leads to the formation of hydroxyl radicals.

### 3.2. Conventional Electrocoagulation Method

3.2.1. Effect of Initial pH

PH is considered one of the main operational factors in the EC process. In this experiment, the effect of solution pH was investigated on the efficiency of electrocoagulation method in removing cyanide, copper, zinc, and nickel (Figure 3). As shown in Figure 3, when using conventional electrocoagulation, the concentration of cyanide decreased from 75 to 14 mg/L with increasing pH from 8 to 10, and then the residual concentration of

cyanide increased from 14 to 26 mg/L by increasing pH from 10 to 12. The same trend was observed for the elements copper, zinc, and nickel, so that by increasing the pH from 8 to 10, the residual concentration of copper decreased from 74 to 31, zinc from 98 to 90, and nickel from 56 to 45 mg/L. Then, by increasing pH from 10 to 12, the residual concentrations of these elements increased as follows: copper from 31 to 49, zinc from 90 to 103, and nickel from 45 to 61 mg/L. The highest removal efficiencies obtained for nickel, cyanide, zinc, and copper were 45, 92, 73, and 81% in 60 min at pH 10 with an electric current of 300 mA using Fe-SS as anode-cathode electrode arrangement, respectively. The rate of change in the copper and cyanide removal efficiencies was relatively high, while the rate of change in nickel and zinc removal efficiencies was relatively low.

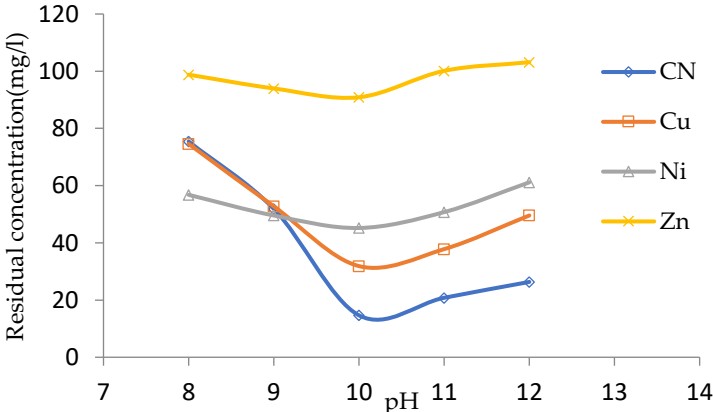

**Figure 3.** Effect of pH on contamination removal efficiency at EC process.

Based on Equations (4) and (5), as the iron anode begins to react and oxidize, the solution pH begins to rise, and hydroxyl ions are produced. Then, based on Equations (6) and (7), cyanide in the solution at high pHs in the presence of hydroxyl ions is oxidized to cyanate ions ($CNO^-$) and then to carbonate or carbon dioxide ($CO_2$) and nitrogen ($N_2$). Consequently, the cyanide ions are removed from the solution by oxidation and production of secondary compounds.

$$Fe_{(s)} \leftrightarrow Fe_{2+(aq)} + 2e^- \tag{4}$$

$$2\,H_2O + 2e^- \leftrightarrow H_{2(g)} + 2OH^- \tag{5}$$

$$CN^- + 2OH^- \leftrightarrow CNO^- + H_2O + 2e \tag{6}$$

$$2\,CNO^- + 4\,OH^- \leftrightarrow 2\,CO_2 + N_2 + 2\,H_2O + 6e \tag{7}$$

Meanwhile, ferric ions react with hydroxyl radicals ($OH^-$) present in the effluent to produce metal hydroxides, including $Fe(OH)_{3(S)}$ or $FeOOH_{(S)}$. These metal hydroxides trap other metal contaminants and heavy metals and eventually precipitate. At low pHs, the rate of OH ion production is low, which results in less contaminant being removed from the effluent. The pH of the solution rises quickly since hydroxyl ions in the solution likely do not all establish bonds with ferric ions. Most contaminants are gradually eliminated from the effluent over time, although hydroxyl ions are still produced. The decline in removal efficiency by increasing pH is due to the production of excessive amounts of hydroxides in the solution because the repulsive force of hydroxides overcomes other forces [27]. Studies showed that the optimum pH for heavy metal removal is mainly in the range of 4–6 [28]. According to the pH activity diagram of the nickel element, $Ni^{2+}$ ions are stable in the pH range of 1 to 8 but, at higher pHs, $Ni^{2+}$ ions are eliminated from the solution by forming metal hydroxides. The stability conditions of zinc, copper, and nickel in aqueous media at the pH values expressed in the Pourbaix diagram are shown in Figure 4.

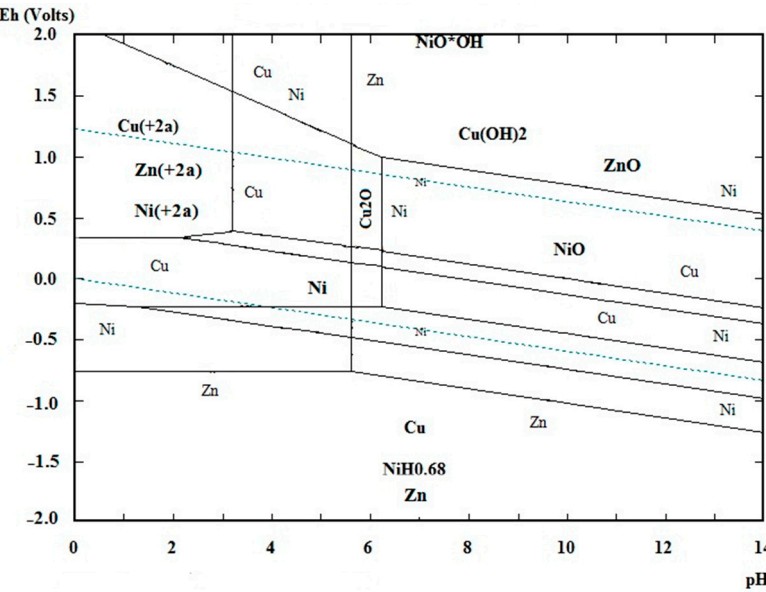

**Figure 4.** Pourbaix diagram of zinc, copper, and nickel metals at 25 °C and 1 atmospheric pressure.

As could be observed in Figure 3, the optimum pH for the elimination of cyanide and heavy metals simultaneously is pH = 10, which causes the elimination of as many contaminants as possible [29]. Consequently, this optimal pH was used in other tests.

### 3.2.2. Effect of Electrode Arrangement

Figure 5 represents the effect of electrodes material on the simultaneous elimination of copper, zinc, nickel, and cyanide. Using a current intensity of 300 mA to the electrodes for 60 min at pH 10 and starting the reaction, the concentration of heavy metal and cyanide ions began to decrease. When using stainless steel as the cathode, the residual concentrations of contaminants were reduced as follows: cyanide from 65 to 8, nickel from 65 to 45, copper from 42 to 21, and zinc from 89 to 28 mg/L. When aluminum was used as the cathode, the residual concentrations of pollutants were reduced as follows: cyanide from 70 to 8, zinc from 85 to 48, nickel from 118 to 60, and copper from 78 to 42 mg/L. According to Figure 5, when using stainless steel as the cathode, the elimination rate of contaminants initially changed slightly but after 20 min it reached a suitable intensity. When aluminum was utilized as the cathode, the same pattern was clearly seen, albeit with a slower rate of change in the pollutant concentration. The most contaminants from the solution were removed simultaneously when the iron electrode served as the sacrificial anode. Some of the reasons for using iron electrodes as the sacrificial anode could be noted as follows: higher oxidation potential and oxidation rate of iron than aluminum (−0.447 v), reasonable price, and ease of access [30]. Recent research showed that aluminum electrodes in the effluent mostly play the role of an amphoteric material [31]. Considering the higher oxidation potential of stainless steel as well as the production of more oxygen and air bubbles compared to aluminum, the removal rate of contaminants is faster when using stainless steel electrodes [32]. According to recent research, the use of stainless steel cathode (SS) leads to the production of larger coagulants compared to coagulants produced using aluminum cathode [33]. According to Figure 5, the highest removal efficiencies of nickel, cyanide, zinc, and copper were 76.4, 98.1, 84.5, and 88.9% using SS-Fe as cathode-anode electrodes, respectively.

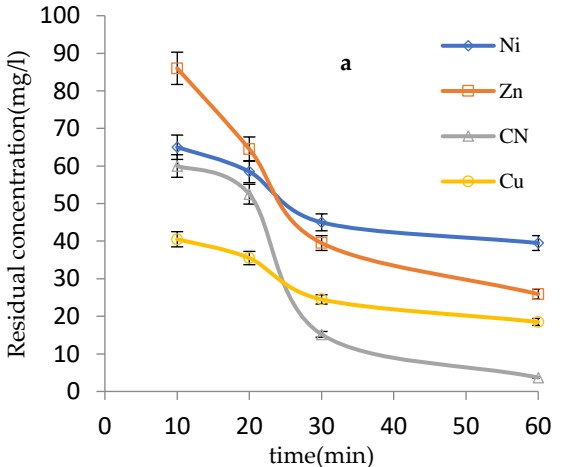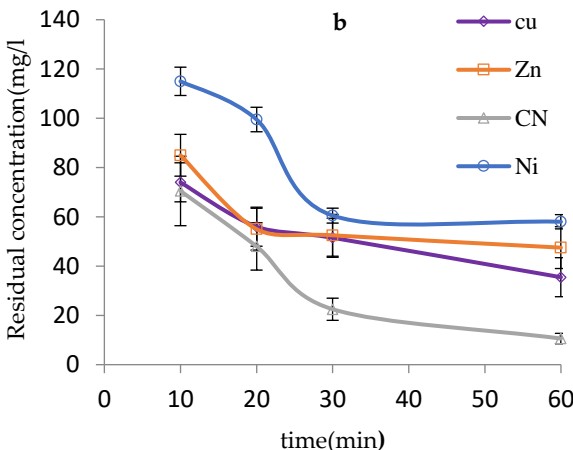

**Figure 5.** Removal of cyanide and heavy metal by EC process. (**a**) Fe-SS; (**b**) Fe-Al.

The process of removing heavy metals and cyanide from the solution using EC technique is shown in Equations (8)–(12). According to Equations (8)–(10), by starting reactions and electrical operations, the iron anode begins to oxidize, and the cathode begins to shrink. Ferrous ions react with cyanide ions to form cyanogenite complexes according to Equations (11)–(13).

$$Fe_{(s)} \leftrightarrow Fe_{2+(aq)} + 2e^- \tag{8}$$

$$H_2O \leftrightarrow 2H^+ + \frac{1}{2}O_2 + 2e^- \tag{9}$$

$$2H_2O + 2e^- \leftrightarrow H_{2(g)} + 2OH^- \tag{10}$$

$$Fe^{2+} + 6CN^- \rightarrow Fe(CN)_6^{4-} \tag{11}$$

$$2Fe^{2+} + Fe(CN)_6^4 \rightarrow Fe_2[Fe(CN)_6] \tag{12}$$

$$Fe_2{}^{II}Fe^{II}(CN)_6 \leftrightarrow Fe_4{}^{III}[Fe^{II}(CN)_6]_3 (\downarrow) + 2Fe^{+2} + e^- \tag{13}$$

Over time, iron ions increase in the effluent, react with cyanide, and are converted to $Fe(CN)_6^{4-}$ and its insoluble and immobilized form, namely ferrocyanide ($Fe_2{}^{II}Fe^{II}(CN)_6$), which is rapidly oxidized to $Fe_4{}^{III}[Fe^{II}(CN))_6]_3$ as an insoluble compound [34]. Thus, pollutants in the effluent are removed by adsorption into this complex and cyanide-hydroxide complexes.

### 3.3. Photo-Electrocoagulation Method

To study the performance of advanced oxidation method (photo-oxidation) in combination with EC in the simultaneous elimination of nickel, cyanide, zinc, and copper, the effects of UV and LED were each first studied separately. To investigate the influence of photo-oxidant type on the elimination of nickel, cyanide, zinc, and copper contaminants, LED was used at pH 10 for 60 min. As shown in Figure 6a, when LEDs were used as photo-catalysts, the rate of change in cyanide, copper, and zinc concentrations was high, while the rate of change in nickel concentration was relatively low. By performing the test for 60 min, the concentrations of cyanide and heavy metals were reduced as follows: cyanide from 124 to 68, nickel from 149 to 114, zinc from 162 to 88, and copper from 158 to 90 mg/L. Similar to the LED photo-catalysts figure, Figure 6b demonstrates that when UVC photo-catalysts were employed to concurrently remove pollutants from the solution, the rate of concentration changes was suitable at first, but gradually reduced and concluded with a gentle slope over time. When using UVC in the test for 60 min, the concentrations of cyanide and heavy metals were reduced as follows: cyanide from 62 to 12, nickel from 125 to 88, zinc from 89 to 38, and copper from 98 to 42 mg/L. As shown in Figure 6a,b, the effect of UVC on the elimination of nickel, cyanide, zinc, and copper was more than that of

LED, and their removal speed was faster when using UVC than LED. It was found that, when using LED, the highest removal efficiencies of nickel, cyanide, zinc, and copper were 31, 65, 46, and 45%, respectively, and when using UVC the highest removal efficiencies of nickel, cyanide, zinc, and copper were 77, 93.7, 47.4, and 75.2%, respectively. By the onset of UV radiation, hydroxyl radicals as oxidizing agents are produced, and cyanide oxidation leads to the production of cyanate ($OCN^-$) as an intermediate product, which is attributed to the presence of superoxide radicals according to Equations (14) and (15). The intermediate product is re-oxidized by hydroxyl and superoxide ($O_2$) radicals to form $NO_2$, $NO_3$, $N_{2(g)}$, and carbonate species, including $H_2 CO_3$, $HCO_3$, $CO_3^2$, and $CO_{2(g)}$. Furthermore, the formation of the final products leads to further oxidation of CN and the formation of the intermediate product ($OCN^-$) [12] according to Equations (16) and (17).

$$H_2O_2 + hv \rightarrow 2OH \tag{14}$$

$$CN^- + 2OH \rightarrow OCN^- + H_2O \tag{15}$$

$$OCN^- + 3OH \rightarrow HCO_3^- + \frac{1}{2}N_2 + H_2O \tag{16}$$

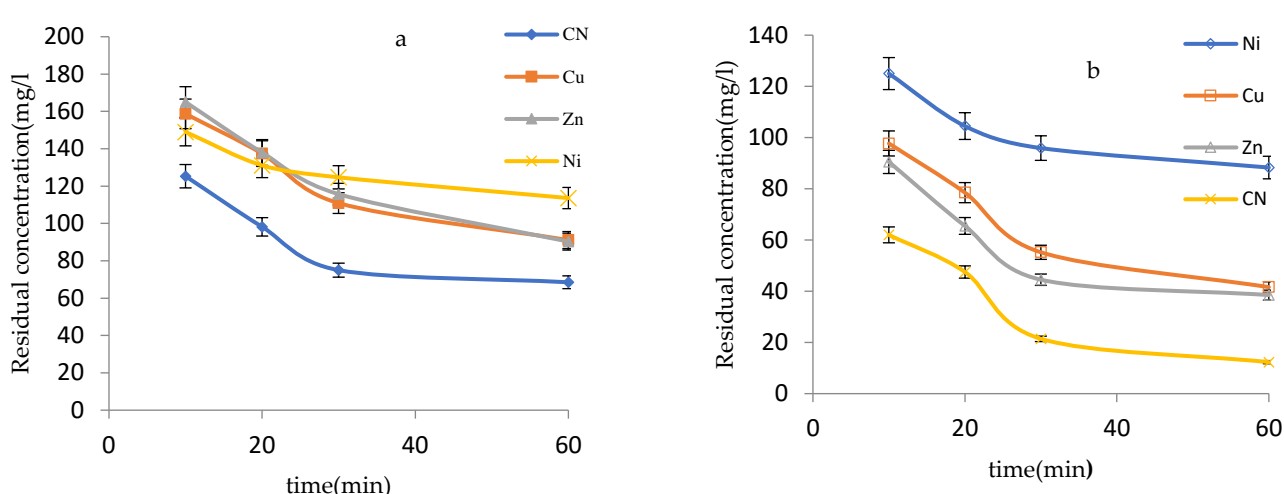

**Figure 6.** Removal of pollutions using (**a**) LED, (**b**) UVC without EC process.

Finally, cyanate is oxidized to bicarbonate and nitrogen [12].

By the onset of UV radiation and the production of hydroxyl ions, copper, nickel, and zinc ions react with hydroxyl ions and oxidize according to Equation (17).

$$Cu^+ + 2OH^- + hv \rightarrow CuO + e^- + H_2O \tag{17}$$

### 3.3.1. Effect of Electrode Arrangement

To evaluate the influence of electrodes material on the elimination of copper, zinc, nickel, and cyanide, Al and SS electrodes were used as cathodes. Based on the past literature indicating the effectiveness of iron electrodes in removing contaminants, iron electrodes were used as the sacrificial anode in all tests, and acidic pH was considered as the most suitable pH for removing heavy metals [29]. As shown in Figure 7, when the aluminum electrode was used as the cathode, the pollutants' concentration decreased rapidly and steeply, stabilizing after 40 min and becoming comparatively uniform. When using the stainless-steel electrode, the concentration of the remaining contaminants was relatively low, and the removal rate of contaminants using an electric current to the electrodes was uniform. The rate of change in zinc, cyanide, and copper concentrations was from 56.82 to 8.52, 14 to 7, and 25.37 to 0.3 mg/L when using the aluminum cathode electrode as well as from 13.38 to 3.7, 8 to 3, and 5.33 to 0.121 mg/L when using the stainless-steel electrode, respectively. According to Equation (18), with the onset of the

EC reaction, the iron electrode as the sacrificial anode on the opposite side of the cathode begins to oxidize [35], and Fe is oxidized to $Fe^{2+}$, and more iron ions are produced with further oxidation of iron. The hydroxyl ions in the effluent then react with the iron ions to form iron hydroxide/polyoxyhydroxide/polyhydroxide flocs [36]. These flocs then adsorb nickel, cyanide, zinc, and copper to form a complex causing the flocs to deposit based on Equations (19)–(23). The adsorption processes in this case include complexation, coagulation, deposition, and neutralization of surface load.

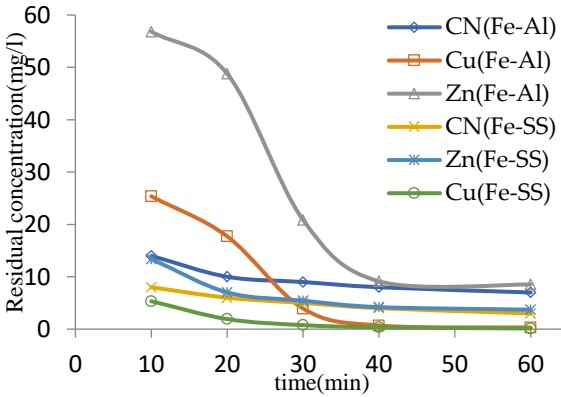

**Figure 7.** Removal of pollutions by $(PEC)_M$ using Fe-Al and Fe-SS.

At the anode surface:

$$4Fe_{(s)} \rightarrow 4Fe_{2+ \ (aq)} + 8e^- \tag{18}$$

At the cathode surface:

$$3H_2O + 3\,e^- \rightarrow 3/2H_{2(g)} + 3OH^- \tag{19}$$

$$Fe^{2+} + 6CN^- \rightarrow Fe(CN)_6{}^{3-} \tag{20}$$

$$4Fe^{2+} + 3Fe\,(CN)_6{}^{3-} \rightarrow Fe_4[Fe(CN)_6]_3 \tag{21}$$

In general:

$$FeOOH_{(s)} + Ox_{(aq)} \rightarrow [FeOOH-Ox] \text{ (adsorption)} \tag{22}$$

$$nFex(OH)(3\,x-y)\,y + Ox_{(aq)} \rightarrow \{\,[Fex(OH)(3\,x-y)\,y\,]_n \cdot Ox\}_{(s)} \text{ (co-precipitation)} \tag{23}$$

Zinc, nickel, copper, and their cyanide complexes are easily adsorbed and precipitated by iron hydroxides. The following equations represent the deposition and removal of zinc from the effluent [35].

$$Zn^{2+} + nCN^- \rightarrow Zn(CN)_n{}^{-(n-2)} \tag{24}$$

$$Fe(OH)_{2(s)} + Zn(CN)_n{}^{2-n} \rightarrow [Fe(OH)_2.\,Zn(CN)_{n(s)}{}^{2-n}]_{(s)} \tag{25}$$

$$Fe(OH)_{3(s)} + Zn(CN)_n{}^{2-n} \rightarrow [Fe(OH)_3.\,Zn(CN)_{n(s)}{}^{2-n}]_{(s)} \tag{26}$$

$$Fe(OH)_{3(s)} + Cu(CN)_n{}^{2-n} \rightarrow [Fe(OH)_3.\,Cu(CN)_{n(s)}{}^{2-n}]_{(s)} \tag{27}$$

$$Fe(OH)_{3(s)} + Ni(CN)_n{}^{2-n} \rightarrow [Fe(OH)_3.\,Ni(CN)_{n(s)}{}^{2-n}]_{(s)} \tag{28}$$

By the onset of reaction and oxidation of the iron anode, a variety of iron hydroxide/oxide species are produced, which adsorb and deposit pollutants. The most stable types of iron hydroxides formed include goethite, lipidocyte, or hematite [37]. According to Equation (24), cyanide ions in the effluent react with metal ions to form cyanide-metal complexes. By producing hydroxyl ions and then forming iron hydroxides, metal-cyanide complexes are adsorbed to these flocs, causing the flocs to precipitate according to Equations (25)–(28).

Based on a study by Taqvi et al. (2008), the dominant species of zinc at pH = 8 based on their percentages are $Zn^{2+}$ (90%), $Zn(OH)_2$ (5%), and $Zn(OH)^+$ (5%), while the

common species of zinc and their percentages at pH = 9 are $Zn(OH)_2$ (78%), $Zn^{2+}$ (13%), and $Zn(OH)^+$ (9%), respectively. Furthermore, the predominant species of zinc at pH = 10 include $Zn(OH)_2$ (93%) and $Zn(OH)_3^-$ (5%), respectively [38].

As mentioned earlier, in terms of the special conditions of stainless steel electrodes compared to aluminum, such as higher oxidation potential, as well as larger coagulant production, the removal of more contaminants using stainless steel was not unexpected. According to Figure 7, the highest removal efficiencies of nickel, copper, zinc, and cyanide were 78.9, 89, 91.2, and 94% when using iron-stainless steel electrodes, respectively.

### 3.3.2. Effect of Initial pH

The findings of this experiment are shown in Figure 8. As shown in Figure 8, when using photo-electrocoagulation, the cyanide concentration decreased from 62 to 8 mg/L with an increase in the pH from 8 to 10, and then the residual concentration of cyanide increased from 8 to 20 mg/L by increasing pH from 10–12. The same trend was observed for the elements copper, zinc, and nickel, so that by increasing the pH from 8 to 10, the residual concentrations of these elements decreased as follows: copper from 57 to 21, zinc from 78 to 67, and nickel from 52 to 22 mg/L. Then, with an increase in pH from 10 to 12, the residual concentrations of these metals increased as follows: nickel from 22 to 44, zinc from 67 to 86, and copper from 21 to 35 mg/L, respectively. With Fe-SS as the anode and cathode electrode configuration, the maximum removal efficiencies of nickel, cyanide, zinc, and copper were 48, 89, 73, and 78% in 60 min, respectively. While the rate of change in the concentrations of cyanide, zinc, and copper was rather significant, that of nickel was comparatively modest. Using an electric current to the electrodes in the EC method, the sacrificial anode, which is mainly made of iron, is oxidized, and iron ions ($Fe^{2+}$) enter the effluent. Simultaneously, hydroxyl and $O_2$ ions are produced, which cause the oxidation of $Fe^{2+}$ to $Fe^{3+}$; on the other hand, cyanide in the effluent is oxidized to cyanate ions ($CNO^-$) at high pHs and then to carbonate or $CO_2$ and $N_2$. At lower pHs, the oxidation rate of ferrous ions ($Fe^{+2}$) to ferric ions ($Fe^{+3}$) is lower, thereby slowing down the removal rate of contaminants. By the onset of the reaction and the production of OH ions, the effluent pH increases, which causes the formation of more metal hydroxides and the removal of contaminants. Over time, the amount of pollutants in the effluent decreases, while the production of OH ions continues. This causes the repulsive force of $OH^-$ to overcome other forces and reduce the adsorption of residual contaminants on metal hydroxides.

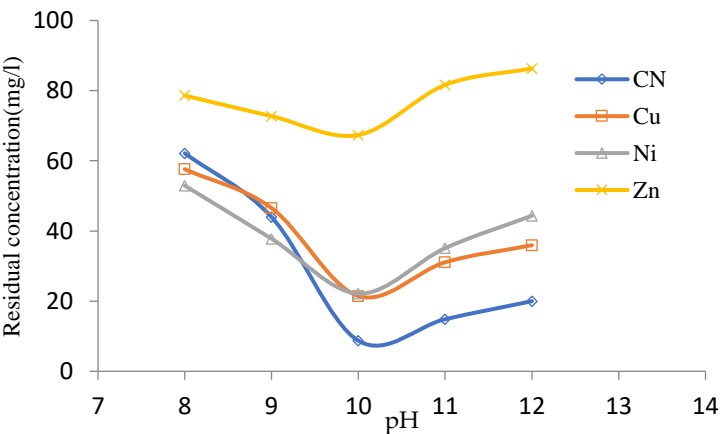

**Figure 8.** Effect of pH on $(PEC)_{PH}$ process for simultaneous removal of pollution.

Copper ions at pH values above 3 are adsorbed to metal-hydroxide coagulants to form metal hydroxide complexes, mainly $Cu(OH)_{2(s)}$. Zinc and nickel ions are oxidized to ZnO and NiO at pH values of about 6 and above and trapped by coagulants dispersed in the solution.

### 3.3.3. Effect of Applied Current Intensity and Reaction Time

As shown in Figure 9a, by the initiation of the experiment using an electric current, the amount of remaining heavy metals and cyanide began to decrease. The quantities of contaminants reduced as follows when an electric current of about 100 mA was employed for 60 min: cyanide from 129 to 54, copper from 138 to 65, zinc from 144 to 73, and nickel from 160 to 122 mg/l. Under the same circumstances, the applied current intensity was increased to 200 mA, which likewise increased the rate at which cyanide and heavy metals were removed. As shown in Figure 9b, using an electric current of 200 mA for 60 min, the concentrations of contaminants decreased as follows: cyanide from 107 to 38, copper from 121 to 54, zinc from 118 to 37, and nickel from 126 to 67 mg/L. By increasing the intensity of applied electric current to 300 mA while maintaining the same conditions in this test, the removal rate of contaminants increased. As shown in Figure 9c, using an electric current of 300 mA for 60 min, the residual concentrations of pollutants decreased as follows: cyanide from 55 to 10, copper from 50 to 28, nickel from 76 to 55, and zinc from 36 to 19 mg/L. As could be observed in Figure 9, the rate of contaminant removal over time was directly associated with the intensity of applied electric current; the effect of changes in the applied current intensity was clear at the beginning of the reaction. The removal efficiencies of nickel, cyanide, zinc, and copper after 60 min were 27, 72.2, 56.5, and 61.3% when applying a current intensity of 100 mA, as well as 60.11, 77.2, 78.5, and 70.2% when applying a current intensity of 200 mA, respectively. According to the results, when the current intensity increased from 100 to 300 mA, the removal efficiencies of nickel, cyanide, zinc, and copper after 60 min were 57, 95, 88.2, and 82.9%, respectively.

According to the test findings and diagrams, the rate at which pollutants are removed from the solution rises as the electric current applied to the electrodes is intensified. The reason for this is that more pollutants are removed because of the greater levels of coagulants produced as a result of the increased current intensity, because the current intensity parameter directly affects the production of coagulants and air bubbles. Furthermore, this parameter is effective in mixing the solution and mass transfer in the electrodes. This parameter also controls the growth of clots and reduces their size, which leads to faster removal and more flotation of contaminants. At the same time, iron hydroxide species increase in the solution and cause other contaminants to be trapped and deposited, which eventually leads to the removal of contaminants from the solution [39]. The cathode used in this test was made of stainless steel. In addition to having a higher oxidation potential than aluminum, it produces more gas and air bubbles as well as larger coagulants, which increase the removal efficiency of pollutants. Based on Pourbaix diagram and the fact that the tests were performed at pH = 10, the predominant species of copper, zinc, and nickel at this pH were mainly NiO, ZnO, and Cu(OH)$_2$, which were formed with the start of the reaction and the production of hydroxides; metal hydroxides mainly Fe(OH)$_3$ adsorb metal oxides and then precipitate.

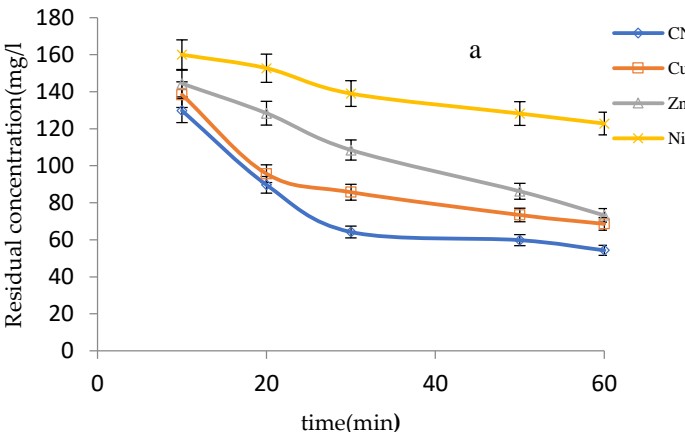

**Figure 9.** *Cont.*

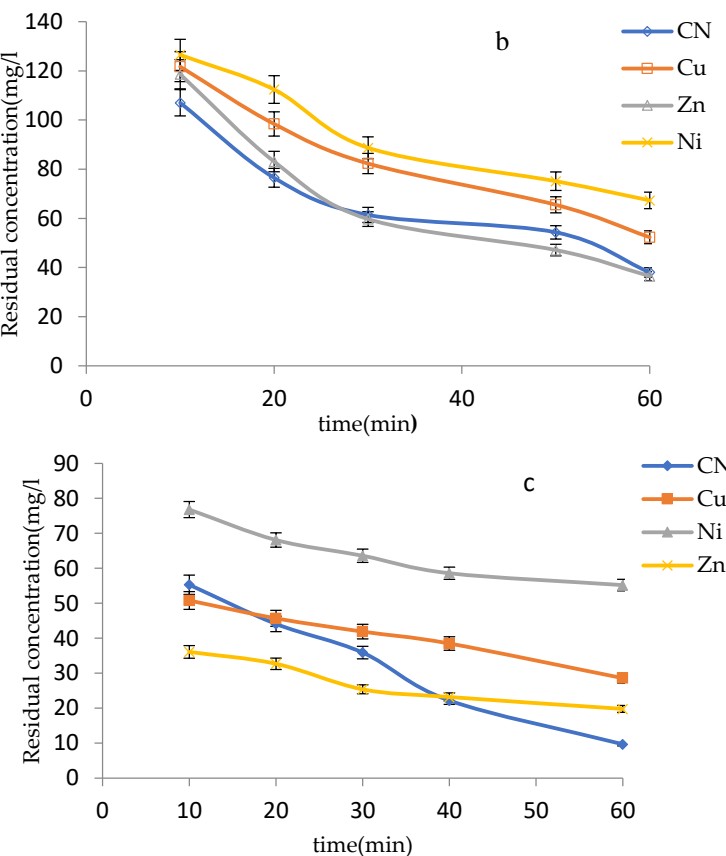

**Figure 9.** Effect of the applied current intensity at (PEC)$_{CI}$ with a current intensity of: (**a**) 100, (**b**) 200, and (**c**) 300 mA.

### 3.4. Effect of Photo-Electrocoagulation-Oxidation Method

As shown in Figure 10, using an electric current to the electrodes and the initiation of the reaction in the presence of photo-catalysts and oxidant, the concentration of heavy metals and cyanide began to decrease so that, after 60 min of purification, the concentration of cyanide was reduced from 28 to 6, nickel from 32 to 25, zinc from 38 to 9, and copper from 31 to 3 mg/L. As shown in Figure 10, at the beginning of the reaction, the rate of change in the concentration of zinc and copper ions was high in the first 20 min, while the rate of change in cyanide concentration was lower than in other heavy metals. In addition, changes in nickel concentration were not significant. Based on Figure 10, copper and zinc in the effluent act as catalysts, while nickel prevents the removal of other metals during the electrocoagulation process.

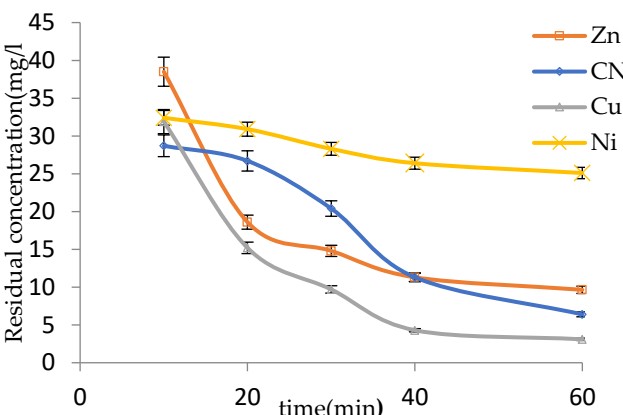

**Figure 10.** The effect of (PECHp) process for simultaneous removal of pollution.

The highest removal efficiencies of nickel, cyanide, zinc, and copper were 85, 96.7, 94.2, and 98% at pH 10 after 60 min of photo-electrocoagulation-oxidation process.

Hydroxyl radicals are produced with the onset of the reaction and UV irradiation according to Equation (29). Based on Equations (30) and (31), by applying an electric current and the oxidation of iron anode, ferrous ions are produced, which react with hydrogen peroxide to form ferric and hydroxyl ions. Some of the ferric ions produced react with hydroxides to form coagulants, and some other ferric ions are reduced by UV radiation and converted to $Fe^{2+}$ ions. The formed iron hydroxides adsorb the pollutants in the effluent and then precipitate.

$$2 \, H_2O + hv \rightarrow 2 \, OH^{\bullet} + H^+ + H + e \tag{29}$$

$$Fe^{2+} + H_2O_2 \rightarrow Fe^{3+} + HO + OH^- \tag{30}$$

$$Fe^{3+} + H_2 O + hv \rightarrow Fe^{2+} + HO + H^+ \tag{31}$$

$$Fe^{3+} + H_2O \rightarrow Fe^{2+} + HO + H^+ \tag{32}$$

The cyanide could be removed in a variety of ways, such as UV radiation, oxidizing hydrogen peroxide, hydroxyl, as well as adsorption to iron hydroxide coagulants. The removal rate of cyanide with a steep slope was not unexpected. The conditions required for copper and zinc removal were similar to those required for cyanide removal. The rate of change in nickel concentration was not relatively large, which could be attributed to the pH used in this test, because the most suitable pH for the removal of nickel is 3 [15].

Furthermore, the presence of cyanide in the solution may lead to the formation of heavy metal-cyanide complexes with different stabilities (nickel-and copper-cyanide as strong complexes, and zinc-cyanide as a weak complex), the formation of strong cyanide-metal complexes in the effluent is one of the reasons for the decrease in heavy metal removal efficiency, which may also be effective in inhibiting cyanide removal. The rate of change in the concentration of pollutants varies, and sometimes the concentration changes are quite slight, due to the chance of cyanide-metal complexes with varying stabilities developing. The cyanide ions in the effluent might combine with copper, silver, gold, tetracyanide, hexacyanide, or nickel (di cyanide). The reaction between free cyanide and ferrous ions to form a complex is described by Equation (33). Iron-cyanide complexes that are not degraded by hydrogen peroxide are stable in effluents, but the reaction with copper ions increases the possibility of forming copper-iron-cyanide complex, which is insoluble and precipitates rapidly [40].

$$2Cu^{2+} + Fe(CN)_6{}^{4-} \rightarrow Cu_2Fe(CN)_6 \text{ (solid)} \tag{33}$$

Some studies showed the catalytic role of copper in the simultaneous elimination of heavy metals from the effluent [40]. The relatively low decrease in nickel concentration compared to other metals may be in terms of the production of nickel-cyanide complexes that are soluble in the effluent, and the possibility of their removal is relatively low due to their relatively high stability.

After obtaining the optimal operating parameters when using synthetic wastewater, to study the efficiency of photo-electrocoagulation-oxidation method on the treatment of gold processing factories' wastewater, real effluent was collected from the gold mines located in Yazd city, and the analysis of the factory effluent is presented in Table 3.

**Table 3.** Analysis of the effluent taken from the gold processing plant.

| CN (ppm) | Ni (ppm) | Zn (ppm) | Cu (ppm) |
|----------|----------|----------|----------|
| 20 | 37.51 | 85.3 | 42.69 |

As shown in Figure 11, by starting the reaction, the concentration of heavy metals and cyanide began to decrease, so that after 60 min of purification, the concentration of cyanide decreased to 2, nickel to 15, zinc to 25, and copper to 9 mg/L.

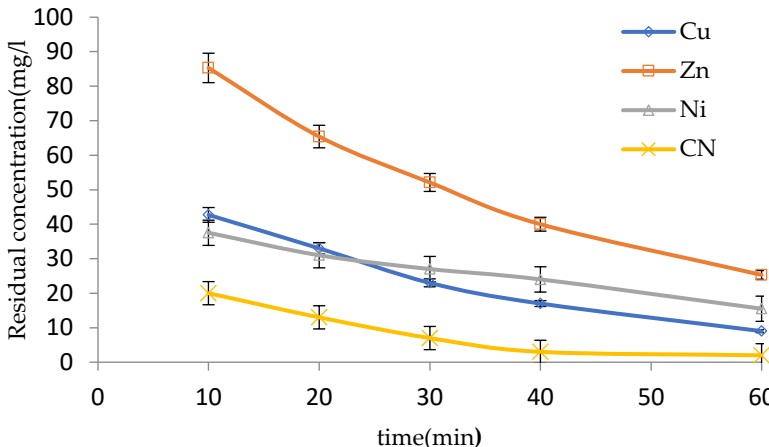

**Figure 11.** The effect of (PECHp) process for simultaneous removal of pollution with real wastewater.

The highest removal efficiency of nickel, cyanide, zinc, and copper was 58, 90, 70, and 78% after 60 min of photo-electrocoagulation-oxidation process. Considering the presence of various types of elements in the effluent of gold processing plant, the possibility of their competition to remove or prevent the removal of each other is high, which is very effective in the removal efficiency of these pollutants.

Figure 12 shows the scanning electron microscopy (SEM) patterns that correspond to the anode surface before and after EC. As shown in Figure 12a, the surface texture of anode electrode before use in EC tests is smooth and relatively uniform. The identical iron electrode's SEM picture after several EC testing is shown in Figure 12b. The iron electrode's smooth surface turns rough and scaly after 60 min of testing. The active locations for the formation of iron hydroxides during oxidation are these bumps. The formation of a large number of these bumps may be related to the presence of nickel, zinc, and copper ions in the effluent. Energy-dispersive analysis of X-rays (EDAX) was used to evaluate the elemental components of zinc, carbon, silica, and nickel adsorbed to iron hydroxides, the results of which are shown in Figure 12. In this figure, the presence of nickel, carbon, silica, zinc, iron, and O is visible in the spectrum. The results of EDAX showed that zinc/nickel was adsorbed onto iron hydroxides. Other detected elements adsorbed to iron hydroxides were chemicals used in the tests and anode/cathode impurities.

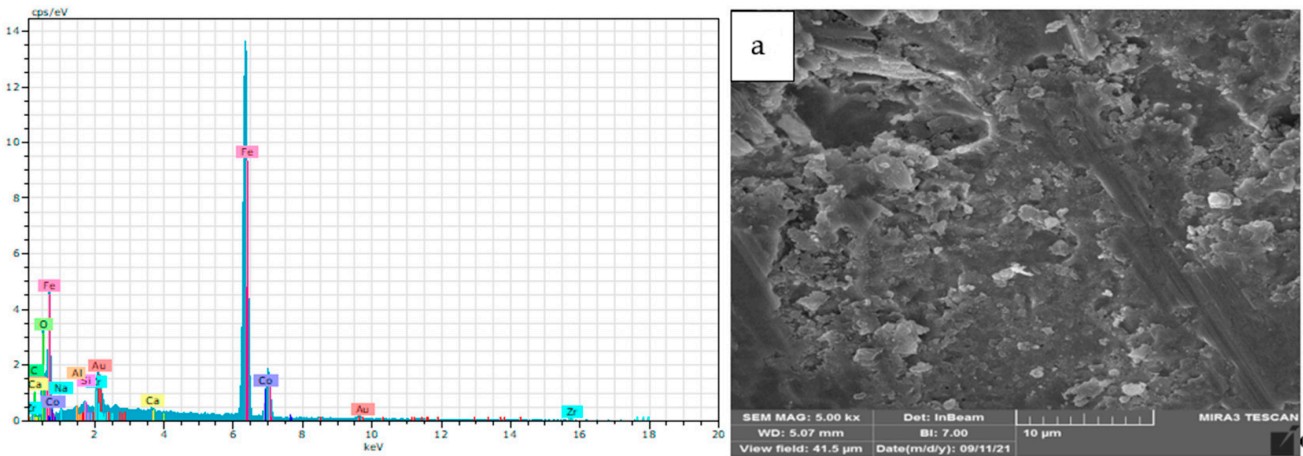

**Figure 12.** *Cont.*

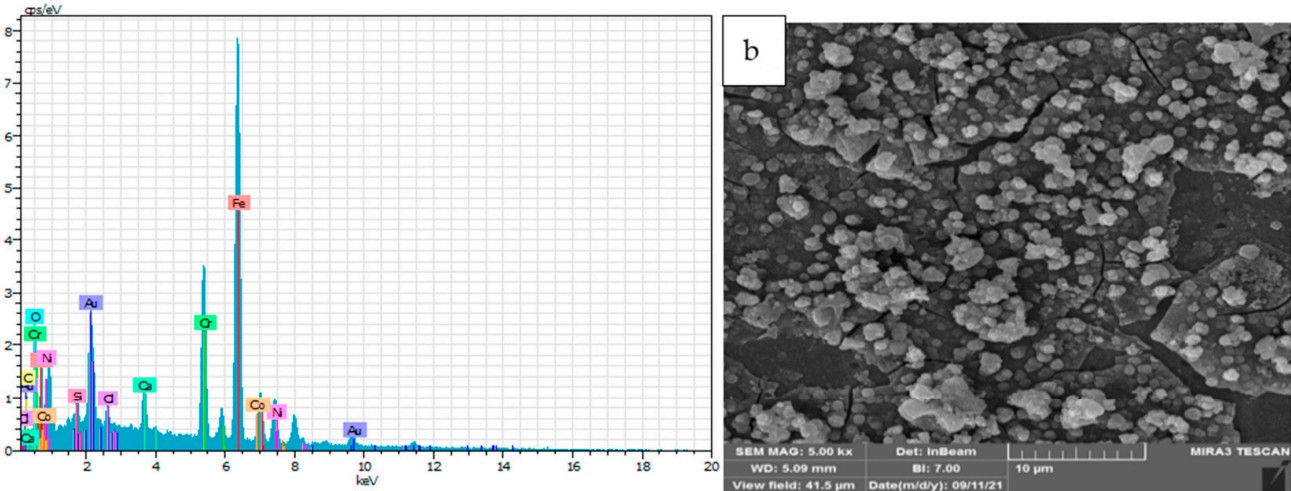

**Figure 12.** SEM images of iron surface before (**a**) and after EC (**b**).

## 4. Energy Consumption and Metal Loading

Energy consumption is a parameter that directly affects operating costs. This parameter depends on several factors that are calculated via the following Equations.

$$E = EC + E\ photo \tag{34}$$

$$EC = \frac{U \times I \times t}{1000 \times V} \tag{35}$$

$$E\ photo = ELED + E\ UV \tag{36}$$

where U represents the voltage applied, I represent the electric current, t is the time (h), and V is the treated effluent volume (m$^3$).

Moreover, the amount of energy consumed in the photo system is calculated using the following Equation:

$$E\ photo = \frac{Pe \times t \times 1000}{V \times 60 \times \log\left(\frac{Ci}{Cf}\right)} \tag{37}$$

where Pe represents the electrical power (kW), t is the time (min), V is the treated effluent volume (m$^3$), Ci is the initial pollutant concentration, and Cf is the final pollutant concentration (mg/L). According to the results, the maximum amounts of energy consumed when using EC, PEC, and PECHp were 8, 22, and 30 kWh/m$^3$, respectively.

One of the common variables in most tests, which indicates the speed of the test process, is the charge load (CL), which is known as the charge density and is calculated in Coulomb/liter (C/L).

$$CL = \frac{I \times tEC}{\nu} \tag{38}$$

In the above equation, I represents the applied current (A), tEC is the EC time (min), and $\nu$ is the solution volume (L or m$^3$) in the EC reactor, respectively. Metal loading (ML) and CL are related to each other based on following equation and Faraday's law:

$$ML = \frac{\Phi \times CL \times MW}{z \times F} \tag{39}$$

In above equation, $\Phi$ repsesents the flow efficiency, MW is the molecular weight of iron, F is Faraday's constant (96485 C/mol e), and z is the number of electrons involved in the oxidation/eduction reaction (z = 2 for iron), respectively (Table 4).

**Table 4.** The effect of metal loading on the removal of pollutants.

| ML(EC)mgFe/l | ML(PEC)mgFe/l | ML(PECHp)mgFe/l |
|:---:|:---:|:---:|
| 0.737 | 0.824 | 0.868 |
| 1.475 | 1.649 | 1.736 |
| 2.213 | 2.474 | 2.604 |

## 5. Comparison with Other Studies

Regarding the simultaneous removal of heavy metals, and cyanide using integrated methods with electrocoagulation, no research has been performed that can compare the results of the research with the results of this research. Dindaş et al. (2020) used the combined method of electrocoagulation-electrofenton and photocatalyst to treat the pharmaceutical wastewater. In order to remove resistant organic pollutants, EC and photocatalyst were used to remove total organic carbon (TOC) from wastewater. Furthermore, the combination of EC+EF, EC+PcO and EF+PcO methods with different operating conditions was investigated. The results showed that the highest amount of removal was first observed in 1 h of EF at a current density of 5 mA/cm$^2$ and the optimal Fe:$H_2 O_2$ molar ratio of 1:10 and then 4 h of PcO using 1.5 g/L $TiO_2$ and 10 mM $H_2 O_2$. The maximum removal rate of pollutants was 64.0% TOC, 70.2% COD and 97.8% BOD5 during the process [41]. Tejera et al. (2021) employed UVA-LED photo-Fenton with coagulation and electrocoagulation to remove contaminants from wastewater. The results showed that combining these techniques produced removal rates for color, COD, and SUVA of over 99%, 89%, and 60%, respectively. The important item was the order of using these coagulant methods in such a way that if coagulant (pH = 5; 2 g L$^{-1}$ $FeCl_3$. 6 H2 O) and then electrocoagulation (pH = 5; 10 mA cm$^{-2}$) led to effluent with pH 6.4, 100 mg/L iron were dissolved, while the first EC electrocoagulation (pH = 4); 10 mA cm$^{-2}$) was followed by CC (pH = 6; 1 g L$^{-1}$ FeCl36 H2 O), resulting in an effluent with a final pH of 3.4 and 210 mg L-1 Fe. The pH used in the electrocoagulation method was 5, the used current density was 10 mA cm$^{-2}$, and the electrodes used were both made of iron [42]. In a research conducted by Jegadeesan et al. (2022) to treat landfill leachate using the integrated solar photo-Fenton electrocoagulation method, the results showed that under the conditions of pH = 7, voltage = 4 V, using Al and Fe electrodes, and a distance between the electrodes of 3 cm, 75 COD and 76 colors were removed. Furthermore, the results showed that if the electrical coagulation process and then the solar photo-Fenton process are carried out under the conditions of pH = 3, H2 O2 = 10 g/L and Fe2 + = 1 g/L, 90% COD and 91% of color are removed [43]. The combined method of UVC-LED/H2 O2/EC was used to simultaneously remove cyanide and metal ions of copper, nickel, and zinc from synthetic wastewater, as well as real wastewater from a gold processing plant. The optimal operating conditions for this test included the current intensity of 300 milliamps, using the iron-stainless steel electrodes and pH 10. These conditions were attained by numerous experiments on synthetic sewage, which were then applied to actual effluent. Nickel, cyanide, zinc, and copper all had removal efficiencies of contaminants of 85, 96, 94, and 98%, respectively, as indicated in Table 5. These results are important due to the fact that, based on the studies conducted, regarding the simultaneous removal of cyanide and heavy metal ions, no study was conducted with the aforementioned methods.

**Table 5.** Studies on the removal of heavy metals with hydrogen peroxide.

| Method | Current Density | Electrode | Pollution | pH | Removal (%) | Ref |
|---|---|---|---|---|---|---|
| UVA-LED/EC/Fenton/Coagulant | 10 mA cm$^{-2}$ | Fe-Fe | color COD SUVA | 4–5 | 99 89 60 | [42] |
| EC/Fenton/Solar | 20 mA cm$^{-2}$ | Fe-Al | COD color | 7 3 | 75 76 ——— 90 91 | [43] |
| electrochemical activation of hydrogen peroxide (EAHP, 91.1 mM) | 45 mA cm$^{-2}$ | Steel-Ti | TOC COD | 11.33 | 100 100 | [44] |
| UV/H2O2(10% $H_2O_2$) | - | - | $Pb^{2+}$ $Cu^{2+}$ | 3 | 60.4 83.1 | [45] |
| $O_3/Fe^{2+}/H_2O_2$ | - | - | Pb, Zn, Mn, Cr (VI) | 5.5 | 98.66% 99.22% 95.11% 94.55% | [46] |
| UV-A with $H_2O_2$(75mg/l) | - | - | As(III) | 7 | 100 | [47] |
| UVsolar/$O_3$/$H_2O_2$ UVsolar/$O_3$/$H_2O_2$/$S_2O_8^{-2}$ | - | - | color COD | - | 56,17 29,77 | [48] |
| EC/photocatalytic ($TiO_2$) | 12 mA cm$^{-2}$ | Fe- Fe | TOC | 8-12 | 74 | [49] |
| UV/$Fe^{2+}$/$H_2O_2$/EC | - | Al-Al | TOC COD | 3 | 73 60 | [50] |
| UVC-LED/$H_2O_2$/EC | 15 mA cm$^{-2}$ | Fe-SS | Ni CN Zn Cu | 10 | 85 96 94 98 | present paper |

## 6. Conclusions

The simultaneous elimination of nickel, cyanide, zinc, and copper from a synthetic effluent was studied using the following methods: oxidation (hydrogen peroxide: Hp), EC, photo-EC, and photo-EC-oxidation (Hp). The results showed that when using hydrogen peroxide, the most suitable pH for the removal of contaminants was pH = 9. Using UV-LED photocatalysts combined with hydrogen peroxide caused more removal of desired pollutants. The reason for this is the production of more hydroxyl radicals and the possibility of more oxidation of pollutants. The most suitable cathode electrode for use in electrocoagulation method was stainless steel electrode. Additionally, the majority of pollutants were eliminated from the wastewater when employing the electrocoagulation technique with hydrogen peroxide at pH = 10. In light of the fact that the pH ranges for removing pollutants were specific, choosing pH = 10 as the ideal pH resulted in the maximum and simultaneous elimination of contaminants. Photo-electrocoagulation method was used, along with hydrogen peroxide at a rate of 4 mg/L, which resulted in removing 96, 98, 85, and 94% of cyanide, copper, nickel, and zinc, respectively. The efficiency of combined photo-electrocoagulation-oxidation (Hp) method increased by 10% compared to the conventional electrocoagulation method. The simultaneous generation of hydroxyl radicals and coagulants as a result of the electrocoagulation process may be the cause of the improvement in efficiency. Moreover, the obtained results confirmed the efficiency of enhanced and combined EC to remove the pollutants in the real wastewater of gold processing plants. The combined method of photo-electrocoagulation-oxidation was not addressed in the previous studies for simultaneous removal of cyanide and heavy metals. Using a combined method has made EC able to remove, as much as possible, the pollutants that compete with each other. Based on the results, when performing the photo-electrocoagulation-oxidation, it seems that copper ion played the role of catalyst and nickel ion played the role of barrier in removing other pollutants. Future research on the elimination of potent metal-cyanide

complexes in batch and continuous modes is advised. The investigation of this technique in respect to other contaminants found in the effluents of the gold processing facility is also advised. The effect of other common oxidants when using electrocoagulation on the simultaneous removal different contaminants should be investigated. In general, the promising results showed that enhanced EC through oxidation and photocatalyst can be used for the treatment of real wastewaters containing various types of contaminants.

**Author Contributions:** A.S.: methodology, writing—original draft preparation, investigation A.K.D.: review and editing, visualization, supervision A.J.-Z.: supervision, conceptualization, methodology, F.T.: conceptualization, methodology, M.H.: review and editing, visualization. All authors have read and agreed to the published version of the manuscript.

**Funding:** This research was supported by the Iran National Science Foundation (INSF) under the Grant No 99029020.

**Institutional Review Board Statement:** Not applicable.

**Informed Consent Statement:** Not applicable.

**Data Availability Statement:** The data presented in this study are available on request from the corresponding author.

**Acknowledgments:** The present study was financially supported by the Iran National Science Foundation (INSF) under the Grant No 99029020.

**Conflicts of Interest:** The authors declare no conflict of interest.

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
