# Peer review of "Simultaneous Removal of Cyanide and Heavy Metals Using Photoelectrocoagulation"

_water, doi:10.3390/w15030581_

Round 1
Reviewer 1 Report
In the manuscript, the authors conducted a systematic investigation of simultaneous removal of cyanide and heavy metals using photoelectrocoagulation. However, some problems have arisen in this manuscript, which still need to be further corrected. To this end, major revisions are required, and the authors should revise their manuscripts based on the following specific comments:
(1) The novelty of this work should be emphasized in the Introduction.
(2) There are many grammatical mistakes in this manuscript, and the linguistic standard of the manuscript requires improvement.
(3) Make sure all abbreviations are written out in full the first time used. This is particularly important in the abstract and the conclusions but work through the entire ms carefully from this perspective.
(4) In this work, only synthetic effluent was used to perform electrocoagulation experiments. It is suggested to add the results of real water experiments.
(5) This work should be compared with recently published at least 8 articles in a scientific table.
(6) The paper published in "Polymers" must explain the significant advances provided in the latest approaches and understanding compared to previous literature, and/or demonstrate convincingly potential in new advancements. The Conclusions of your paper are especially important for this. Therefore, please try to sharpen this further. The optimal Conclusion should include:
* A summary of your key points in the review.
* A highlight of the latest approaches, new concepts, and innovations.
* A summary of key improvements compared to findings in the literature [provide a couple of references to indicate key improvements].
* Your vision for future work.
Author Response
Dear Editor and Reviewers:
Thank you for your thoughtful comments on our manuscript entitled “Simultaneous Removal of Cyanide and Heavy Metals Using Photo Electrocoagulation ". Those comments were valuable and helpful for revising, and improving the manuscript. We carefully studied the comments, and made all suggested changes, which helped improve the manuscript significantly. To facilitate the review process, all revisions were made in red color in the revised manuscript. In the following section, we explained in detail how we responded to each comment by repeating the comment, and then providing a response just below it.
Comments from the editors and reviewers:
Reviewer#1
1) The novelty of this work should be emphasized in introduction.
Response
Thank you for your suggestion. The last part of introduction was revised to address the reviewer comment.
2) There are many grammatical mistakes in this manuscript, and the linguistic standard of the manuscript requires improvement.
Response
Thank you for your attention. The manuscript was revised to address the reviewer comment. The language editing certificate is attached.
3) Make sure all abbreviations are written out in full the first time used. This is particularly important in the abstract and the conclusions, but work via the entire is carefully from this perspective.
Response
Thanks for your appropriate comment. The manuscript was revised to address the reviewer comment.
4) In this study, only synthetic effluent was used to perform electrocoagulation experiments. It is suggested to add the results of real water experiments.
Response
Thanks for your appropriate comment. The optimized enhanced EC was performed on real gold mine effluent. The last part of results and discussion was revised to address reviewer comment.
5) This work should be compared to recently published at least 8 articles in a scientific table.
Response
Thank you for your suggestion. Section 5 entitled “comparison with other studies” was added to the manuscript to address the reviewer comment.
6) provided in the latest approaches, and understanding compared to previous literature, and/or show convincingly potential in new advancements. The conclusions of your paper are especially important for it. Therefore, please try to sharpen this further. The optimal Conclusion should include:
* A summary of your key points in the review.
* A highlight of the latest approaches, new concepts, and innovations.
* A summary of key improvements compared to findings in the literature [provide a couple of references to indicate key improvements].
* Your vision for future work.
Response
Thank you for your suggestion. The conclusion was revised to address the reviewer comment.

Reviewer 2 Report
In this study, simultaneous elimination of nickel, cyanide, zinc, and copper from a synthetic effluent was investigated. This manuscript is well written and can be published after the following revisions.
1. There are grammatical errors in the manuscript, and I advise the author to revise them carefully.
2. Figure 3. Why did the author choose this pH (8-12)?
3. It is suggested that the author choose the research work of the past 3 year for comparison. These two studies are helpful to improve the quality of manuscripts. (10.1021/acsami.1c22035 and 10.1016/j.jhazmat.2021.128062)
4. The quality of the pictures should be further improved.
5. Conclusions: I suggest that the author simplify the conclusion and highlight the innovation of the manuscript.
Author Response
Dear Editor and Reviewers:
Thank you for your thoughtful comments on our manuscript entitled “Simultaneous Removal of Cyanide and Heavy Metals Using Photo Electrocoagulation ". Those comments were valuable and helpful for revising, and improving the manuscript. We carefully studied the comments, and made all suggested changes, which helped improve the manuscript significantly. To facilitate the review process, all revisions were made in red color in the revised manuscript. In the following section, we explained in detail how we responded to each comment by repeating the comment, and then providing a response just below it.
Comments from the editors and reviewers:
Reviewer #2
1) There are grammatical errors in the manuscript, and I advise the author to carefully revise them
Response
Thank you for your suggestion. The manuscript was revised to address the reviewer comment. The language editing certificate is attached.
2) Figure 3. Why did the author choose this pH (8-12)?
Response
Thank you for your attention. The reason to choose this pH was to prevent the production of dangerous HCN gas during acidic pH. Thus, the pH in the range of 8 to 12 was chosen.
3) It is suggested that the author chooses the research work of the past 3 year for comparison. These two studies are helpful to improve the quality of manuscripts. (10.1021/acsami.1c22035 and 10.1016/j.jhazmat.2021.128062)
Response
Thank you for your suggestion. The mentioned papers were added to introduction.
4) The quality of pictures should be improved further.
Response
Thank you for your suggestion. The quality of figures was improved.
5) Conclusions: I suggest that the author simplify the conclusion and highlight the innovation of the manuscript.
Response
Thank you for your suggestion. The conclusion section was revised to address the reviewer comment.

Round 2
Reviewer 1 Report
The revised version can be accepted.
Reviewer 2 Report
Accept